# The Influence of 5-HTTLPR, BDNF Rs6265 and COMT Rs4680 Polymorphisms on Impulsivity in Bipolar Disorder: The Role of Gender

**DOI:** 10.3390/genes13030482

**Published:** 2022-03-09

**Authors:** Andrea Boscutti, Alessandro Pigoni, Giuseppe Delvecchio, Matteo Lazzaretti, Gian Mario Mandolini, Paolo Girardi, Adele Ferro, Michela Sala, Vera Abbiati, Marco Cappucciati, Marcella Bellani, Cinzia Perlini, Maria Gloria Rossetti, Matteo Balestrieri, Giuseppe Damante, Carolina Bonivento, Roberta Rossi, Livio Finos, Alessandro Serretti, Paolo Brambilla

**Affiliations:** 1Department of Pathophysiology and Transplantation, University of Milan, 20122 Milan, Italy; andrea.boscutti@unimi.it; 2Department of Neurosciences and Mental Health, Fondazione IRCCS Ca’ Granda-Ospedale Maggiore Policlinico, 20122 Milan, Italy; ale.pigoni@gmail.com (A.P.); g.delvecchio@hotmail.it (G.D.); matteo.lazzaretti@policlinico.mi.it (M.L.); gianmario.mandolini@gmail.com (G.M.M.); adele.ferro1@gmail.com (A.F.); mariagloria.rossetti@univr.it (M.G.R.); 3Social and Affective Neuroscience Group, MoMiLab, IMT School for Advanced Studies Lucca, 55100 Lucca, Italy; 4Department of Developmental Psychology and Socialization, University of Padua, 35131 Padua, Italy; paolo.girardi@unipd.it (P.G.); livio.finos@unipd.it (L.F.); 5Mental Health Department, Azienda Sanitaria Locale Alessandria, 15121 Alessandria, Italy; michelasala@hotmail.com; 6Department of Brain and Behavioral Sciences, University of Pavia, 27100 Pavia, Italy; vera_abbiati@asst-pavia.it; 7Department of Mental Health and Substance Abuse, Azienda Sanitaria Locale Piacenza, 29121 Piacenza, Italy; m.cappucciati@ausl.pc.it; 8Section of Psychiatry, Department of Neurosciences, Biomedicine and Movement Sciences, University of Verona, 37134 Verona, Italy; marcella.bellani@univr.it; 9Section of Clinical Psychology, Department of Neurosciences, Biomedicine and Movement Sciences, University of Verona, 37134 Verona, Italy; cinzia.perlini@univr.it; 10Psychiatry Unit, Department of Medicine, University of Udine, 33100 Udine, Italy; matteo.balestrieri57@gmail.com; 11Department of Medicine (DAME), University of Udine, 33100 Udine, Italy; giuseppe.damante@uniud.it; 12IRCCS “E. Medea”, Polo Friuli-Venezia Giulia, San Vito al Tagliamento, 33078 Pordenone, Italy; boniventocarolina@gmail.com; 13Psychiatry Unit, IRCCS Istituto Centro San Giovanni di Dio FBF, 25125 Brescia, Italy; rrossi@fatebenefratelli.eu; 14Department of Biomedical and NeuroMotor Sciences, University of Bologna, 40123 Bologna, Italy; alessandro.serretti@unibo.it

**Keywords:** bipolar disorder, impulsivity, BIS-11, polymorphism, 5-HTTLPR, BDNF, COMT

## Abstract

Impulsivity has been proposed as an endophenotype for bipolar disorder (BD); moreover, impulsivity levels have been shown to carry prognostic significance and to be quality-of-life predictors. To date, reports about the genetic determinants of impulsivity in mood disorders are limited, with no studies on BD individuals. Individuals with BD and healthy controls (HC) were recruited in the context of an observational, multisite study (GECOBIP). Subjects were genotyped for three candidate single-nucleotide polymorphisms (SNPs) (5-HTTLPR, *COMT* rs4680, *BDNF* rs6265); impulsivity was measured through the Italian version of the Barratt Impulsiveness Scale (BIS-11). A mixed-effects regression model was built, with BIS scores as dependent variables, genotypes of the three polymorphisms as fixed effects, and centers of enrollment as random effect. Compared to HC, scores for all BIS factors were higher among subjects with euthymic BD (adjusted β for Total BIS score: 5.35, *p* < 0.001). No significant interaction effect was evident between disease status (HC vs. BD) and SNP status for any polymorphism. Considering the whole sample, *BDNF* Met/Met homozygosis was associated with lower BIS scores across all three factors (adjusted β for Total BIS score: −10.2, *p* < 0.001). A significant 5-HTTLPR x gender interaction was found for the SS genotype, associated with higher BIS scores in females only (adjusted β for Total BIS score: 12.0, *p* = 0.001). Finally, *COMT* polymorphism status was not significantly associated with BIS scores. In conclusion, BD diagnosis did not influence the effect on impulsivity scores for any of the three SNPs considered. Only one SNP—the *BDNF* rs6265 Met/Met homozygosis—was independently associated with lower impulsivity scores. The 5-HTTLPR SS genotype was associated with higher impulsivity scores in females only. Further studies adopting genome-wide screening in larger samples are needed to define the genetic basis of impulsivity in BD.

## 1. Introduction

Bipolar disorder (BD) is a severe, chronic, and common mental illness, characterized by episodes of elevated mood (hypomania or mania) alternating with depressive episodes [1]. BD is listed among the top causes of years lost due to disabilities [2] and is associated with a lifetime risk of suicide 20–30 times higher than the general population [3]. While mood changes represent the defining feature of the disease, individuals with BD display various mood-independent trait patterns encompassing temperament, character, and personality [4,5,6,7,8,9].

Impulsivity is one psychological dimension that has been shown to be significantly represented in individuals with BD [10,11,12], even during a phase of euthymia [13,14]; moreover, trait impulsivity is increased in unaffected relatives of individuals with BD [15] and in high-risk subjects [16]. As such, impulsivity has been proposed as an endophenotype for BD [17]. Impulsivity is defined as “a predisposition towards rapid, unplanned reactions to internal or external stimuli without regard to the negative consequences of these reactions to the impulsive individual or others” [18] and in BD individuals is associated with at-risk behaviors, such as suicidality, substance abuse and criminal actions [11,19]; it is also a predictor of quality of life [20], disease onset [21] and illness severity [13].

Importantly, impulsivity has a significant genetic background [22] and similar to other complex phenotypes, a great number of genetic variants seem to influence this trait, with small individual effects [23,24]. This may stem from impulsivity being a complex, multidimensional trait, which can be measured through a multitude of instruments (i.e., neuropsychological paradigms, other- or self-administered questionnaires), or through other surrogate measures (i.e., violent, aggressive, or suicidal behavior). Since impulsivity is considered a candidate endophenotype for BD, it may be hypothesized that individuals with BD carry an increased genetic load of impulsivity. However, to date, studies that explored this hypothesis are limited to single candidate genes [25,26,27].

The purpose of the present study is to elucidate the genetic basis of impulsivity in individuals with BD. In doing so, we focused on three single-nucleotide polymorphisms (SNPs)—*COMT* rs4680, 5-HTTLPR (*SLC6A4*) and *BDNF* rs6265. The choice of these three candidate genes was motivated by the following reasons: (1) all three SNPs have been associated with increased susceptibility for BD [28,29,30]; (2) they have been consistently shown to influence behavioral traits in both healthy subjects and individuals with psychiatric disorders (see below for more details). Indeed, the influence of these three SNPs on temperament and character dimensions in both HC and BD patients was investigated by our consortium (GECOBIP) and reported in a separate paper [31]. (3) Finally, all three SNPs have been shown to influence impulsivity measures in samples of HC and neuropsychiatric populations (see below for more details).

The *COMT* gene encodes for an enzyme involved in the regulation of dopamine levels, responsible for about 50% of dopamine clearance in the prefrontal cortex [32]. A common *COMT* polymorphism is a valine to methionine substitution at codon 158 (rs4680) [33] that is associated with markedly reduced enzyme activity [34]. *COMT* polymorphisms have been reported to influence temperament, personality, and cognition [35,36,37,38].

Several studies report an association between the Met substitution and increased impulsivity in healthy individuals [39] and among various neuropsychiatric disorders, including attention deficit hyperactivity disorder (ADHD) [40], binge-eating disorder [41] and borderline personality disorder [42]. To our knowledge, to date no studies on the effect of *COMT* polymorphism on impulsivity in BD are available.

The *SLC6A4* gene encodes for the serotonin transporter (SERT or 5-HTT) protein [43] responsible for the uptake of serotonin from the synaptic cleft [44]. The 5-HTT gene-linked polymorphic region (5-HTTLPR) is a widely studied polymorphism that involves the transcriptional control region upstream of the *SLC6A4* coding sequence [45]. The long and short variants of this region are associated with higher and lower transcriptional activity, respectively [45]. The polymorphism has been found to influence affective temperament [46], and to be associated with higher levels of negative emotionality [47], neuroticism [48], rumination [49], and hostility [50]. Of note, surrogate measures of impulsivity, such as response inhibition, gambling, or attentional tasks, have been shown to be heterogeneously influenced by the S allele in samples of healthy individuals [51,52]. Regarding the relationship between the 5-HTTLPR polymorphism and impulsivity in individuals with BD, increased ADHD-related impulsivity symptoms during childhood were found in a group of individuals with bipolar disorder type 2 [53]. Conversely, no association between HTTPLR status and impulsivity was found in a group of BD individuals with a history of suicide attempt [26].

The *BDNF* gene encodes for a member of the neurotrophin family, a class of proteins involved in the development, survival, and synaptic plasticity of neurons [54]. A valine to methionine substitution at codon 66 (rs6265) is the product of a common polymorphism in the 5′ pro-BDNF region. This substitution was shown to impair intracellular processing and secretion of the mature protein [55]. This SNP has been associated with several traits, such as sensation seeking [56] and harm avoidance [57]. The polymorphism has also been suggested to modulate various cognitive abilities in both healthy individuals and psychiatric populations, with Met substitution generally associated with worse cognitive performances [58,59,60]. The influence of the *BDNF* genotype on impulsivity-related domains appears not well defined, with conflicting results regarding risk of suicidal behavior [61,62,63] and substance abuse/dependence [64,65,66]. We found no reports on the effect of this SNP on impulsivity in samples of BD individuals.

In conclusion, the evidence on the genetic basis of impulsivity in individuals with BD is extremely limited. The present study aims to address this topic by investigating the effect of three functional genetic variants (5-HTTLPR, *COMT* rs4680, and *BDNF* rs6265) on impulsivity, as measured through the Barratt Impulsiveness Scale (BIS-11), in a sample of healthy controls (HC) and BD patients.

## 2. Materials and Methods

### 2.1. Study Sample

Subjects were recruited from 8 Italian clinical sites part of the GECOBIP consortium (Table 1). GECOPIB was built with the aim to explore the association between three SNPs and neuropsychological and clinical characteristics of individuals with BD.

Exclusion criteria for both HC and BD included: (1) age ≤18 years, (2) intelligence quotient ≤70, (3) medical or neurological comorbidities, (4) not being an Italian native speaker. Additional exclusion criteria for HC were a diagnosis of cognitive impairment or an history of alcohol and/or drug abuse. BD diagnosis had to be confirmed through the Structured Clinical Interview for DSM-IV TR axis I (SCID-I); individuals with BD diagnosed with any other axis I psychiatric comorbidity were excluded.

The study was approved by the local Ethics Committee and conducted according to the ethical principles stated in the Declaration of Helsinki.

### 2.2. Assessment Tools

Individuals were assessed for socioeconomical, neuropsychological and clinical characteristics, as reported in Porcelli et al. [31]

In the present study, we focused on the influence of selected genetic variants on the temperamental features of the sample, and in particular on the impulsiveness as measured by the Barratt Impulsivity Scale [67], Italian version [68]. BIS-11 is a self-report questionnaire composed of 30 items; every single item is answered on a 4-point scale (Rarely/Never, Occasionally, Often, Almost Always/Always). Each item contributes to define six first-order factors, which combined produce the following three second-order factors: “Attentional Impulsiveness”, “Motor Impulsiveness” and “Non-Planning Impulsiveness”.

### 2.3. Genotype Analysis

Three functional genetic variants within key candidate genes (*COMT*, *BDNF*, and *SLC6A4*) were genotyped in each individual recruited: the 5-HTTLPR variants within the *SLC6A4* gene, the rs4680 within the *COMT* gene, and the rs6265 within the *BDNF* gene. Genomic DNA was collected from a saliva sample according to standard procedures (Oragene DNA Self-Collection Kit [69]). The *COMT* rs4680, the *BDNF* rs6265, and the 5-HTTLPR polymorphisms were analyzed by real-time polymerase chain reaction allelic discrimination assay, using predesigned TaqMan genotyping assays and standard assay conditions. Personnel performing the genetic analysis were blind to the socioeconomical, neuropsychological and clinical characteristics of the subjects.

### 2.4. Statistical Analysis

Data were summarized with frequency tables and figures. Between-group differences in the distribution of categorical and continuous variables were tested by means of Chi-squared test and Student’s T test, respectively. Since marginal BIS score showed a non-normal distribution (Appendix A), comparisons between HC and BD and among gene polymorphisms were performed by means of Mann–Whitney (MW) test, or Kruskal–Wallis (KW) test if the strata were more than two.

A series of mixed-effects regression models were estimated, with BIS scores (Attentional, Motor, Non-Planning, Total Score) as dependent variables; disease status (HC or BD) and genotypes of the three polymorphisms (*COMT* rs4680, *BDNF* rs6265, 5-HTTLPR) were included as fixed effects. The estimated models considered the presence of a random effect (random intercept) represented by the center of enrollment. Results were presented by coefficients (β) adjusted for a series of confounders: age class (3 tertiles: (18–30.7), (30.7–47) and (47–73) years), gender, Raven IQ score (3 tertiles: (66–114), (114–127), and (127–128) points), educational level (3 tertiles: (5–13), (13–18), and (18–26) years). Interactions between gene polymorphisms and covariates were tested and included minimizing the Akaike’s Information Criterion (AIC) index.

The normality of the regression residuals was verified to evaluate the presence of outliers and deviations from the normal distribution. The observed power analysis and the replication attempt was assessed by means of a Retrospective Design Analysis (RDA) [70]. RDA was applied on the predicted BIS scores of the estimated mixed-effects regression models; this allowed to calculate, for each model and polymorphisms, the retrospective power of the highest observed effect size (Cohen d) among all possible group combinations, and the relative Type M and Type S errors.

The significance was set at 5%. All statistical analysis was performed using R Statistical Software; packages lme4 and lmerTest were used for the estimation of the mixed-effects regression model, while the PRDA package was adopted for the RDA [71].

## 3. Results

### 3.1. Characteristics of the Sample

Overall, 228 subjects were included in the analysis (132 HC, 96 BD). Three subjects with overall BIS score less than 35 were excluded; 225 subjects were considered for the subsequent analysis (131 HC, 94 BD). The sociodemographic characteristics and the results of the psychometric assessments are presented in Table 1. In general, individuals with BD were older (mean (standard deviation)—HC: 35.5 years (12.9); BD 45.7 (11.7); *T*-test *p* < 0.001), less educated (HC: 16.9 schooling years (3.8); BD: 12.7 (4.0); *T*-test *p* < 0.001), and had a lower socioeconomic status as measured by the Barratt Simplified Measure of Social Status (BSMSS) (HC: 44.5 (12.9); BD: 31.2 (15.7); *T*-test *p* < 0.001). The BD group performed significantly worse on the Raven IQ test (HC: 122.7 (7.9); BD: 112.5 (12.4); *T*-test *p* < 0.001).

### 3.2. Impulsivity Scores

BD scored higher on all BIS-11 factors (Attentional, Motor, and Non-Planning) and overall score (HC: 59.0 (12.0); BD: 68.5 (12.8); MW *p* < 0.001) (Table 2 and Appendix A). The prevalence of the genotypes for the three polymorphisms studied did not differ significantly between the two groups (Table 2). Moreover, allelic and genotypic frequencies were consistent with those reported for the Caucasian ethnicity [72,73,74]. Of note, only 14 subjects (6.2%) carried the *BDNF* TT genotype. There were no significant differences in BIS scores between different *COMT* genotypes (Figure 1); conversely, significant differences in BIS scores were found for the 5-HTTLPR and *BDNF* polymorphisms. Specifically, BIS Motor factor and overall scores were lower for BD individuals carrying the 5-HTTLPR L allele (genotypes LL or LS) (KS test < 0.05) (Figure 2). Among both HC and BD subjects, *BDNF* TT homozygosis was associated with lower Non-Planning and overall BIS scores (KS test *p* < 0.05; Figure 3).

### 3.3. Mixed-Effects Regression Model

A total of four regression models were estimated. The β estimates for the BIS Total score are shown in Table 3; the results of the mixed-effect regression models for each BIS factor are reported in Appendix A. The fitted models explained only a discrete quote of variance (marginal R^2^ comprised between 23% and 33.1%). The quantile-quantile diagram reported a good adaption of the regression residuals to the normal distribution, with some deviation only for the Attentional BIS dimension (Appendix A).

#### 3.3.1. Main Effects

Total BIS scores were significantly influenced by gender, with female sex associated with lower BIS scores; there was no association between BIS scores and age. Higher educational level, but not higher Raven IQ, was associated with lower Total BIS scores. BD diagnosis was associated with higher impulsivity scores across all BIS factors.

The *BDNF* rs6265 Met/Met homozygosis was the genotype most consistently associated with lower impulsivity levels, being associated with both lower Total BIS scores and lower scores across each BIS factor when compared to the Val/Val genotype. BIS scores did not differ between the Val/Val homozygosis and the Val/Met heterozygosis. Regarding the 5-HTTLPR polymorphism, LL homozygosis was associated with lower BIS scores in the Non-Planning factor if compared to the LS heterozygosis, but not with lower Total BIS scores. The SS genotype had no effect on impulsivity scores. Finally, *COMT* rs4680 genotype status did not influence any BIS scores.

#### 3.3.2. Interaction Effects

Of note, no significant interaction effect on impulsivity levels was evident between disease status (HC vs. BD) and any polymorphism status, apparently implying no differences in the modulatory effect of the three SNPs on impulsivity between HC and BD individuals. The most consistent interaction found was that between the 5-HTTLPR SS genotype status and gender. In females the SS homozygosis was associated with higher BIS scores across all the three BIS factors when compared to the LS heterozygosis (Figure 4). Of note, no main effect of the SS genotype was evident. Two other significant interactions were found: higher Motor BIS score for the combination *BDNF* TC and age class (31.7–47) and elevated BIS Non-Planning scores for high education levels with GG genotype in *COMT*. However, in both cases the effect was limited to only one BIS factor and did not influence Total BIS scores (Figure 4; Table 3).

No other significant interaction effect was found.

### 3.4. Retrospective Design Analysis

The results of the RDA on the maximum observed effects size between pairwise comparisons are reported in the (Appendix A). Overall, Type S error was small, indicating a high probability of a correct direction of the measured differences. Despite being significant, Type M error was generally small for BIS factor and overall scores for the interaction between the 5-HTTLPR polymorphism and gender. Type M error values were higher for the three BIS factor scores for both *BDNF* and *COMT* polymorphisms. In summary, the RDA confirmed that all the observed results were supported by an adequate power and a low Type M error.

## 4. Discussion

The main findings of the study in our sample of euthymic BD individuals and HC can be summarized as follows:(1)In our sample of euthymic BD individuals, BD diagnosis was associated with higher impulsivity scores across all BIS factors when compared to HC; this finding confirms earlier reports [75], ultimately suggesting that impulsivity may represent a state-independent trait of the disease.(2)The effect of the interaction between BD diagnosis and polymorphism status on impulsivity levels was not significant for any of the three polymorphisms considered.(3)In the whole sample (HC+BD), *BDNF* Met/Met homozygosis (but not Val/Met heterozygosis) was associated with lower BIS scores when compared to the wild-type genotype (Val/Val).(4)A genotype × gender interaction was evident for the 5-HTTLPR polymorphism in females, with the SS genotype (but not the LS heterozygosis) being associated with higher BIS scores. Conversely, no main effect was evident for the SS homozygosis.(5)No significant association was found between *COMT* genotype status and impulsivity levels.

### 4.1. No Differences in the Effect of SNPs on Impulsivity between HC and BD

One of the primary goals of our study was to determine whether the three polymorphisms differed in the two samples in terms of allele frequencies or effect on impulsivity levels. This is a relevant hypothesis to be tested since, if confirmed, it supports the role of impulsivity as an endophenotype for BD [76,77]. However, neither two were confirmed in our sample. Specifically, allele frequencies were comparable between HC and BD individuals. Moreover, the polymorphism status effect on impulsivity scores was not dependent on diagnosis, implying similar modulatory effects on impulsivity in HC and BD individuals. Previous evidence on the role of these three SNPs in modulating impulsivity in BD subjects is limited to two studies on 5-HTTLPR. One found a differential effect of SNP between BD and HC samples; however, the study was focused on ADHD impulsivity features retrospectively measured [53]. The other study, in which cognitive impulsivity was measured through neuropsychological tools, found no differences in the effect of the SNP genotype status on impulsivity between two small samples of HC and BD individuals (non-significant effect in both) [52]. Therefore, since other genes might influence the complex trait of impulsivity and might interact with the diagnosis of BD, genome-wide screening studies on larger samples, with homogeneous impulsivity measures, are needed.

### 4.2. The Effect of BDNF on BIS Scores

In our sample the *BDNF* rs6265 genotype associated with low *BDNF* transcriptional efficiency (Met/Met) was associated with reduced impulsivity levels. While the Met substitution is traditionally associated with low transcriptional efficiency, studies in both healthy subjects [78] and individuals with BD [79] suggest a lack of association between genotype status and circulating BDNF plasma levels. Nonetheless, several studies point towards increased impulsivity levels in individuals carrying the Met allele. Specifically, in healthy individuals Met substitution is associated with worse performances in Go/No Go and Stop Signal Tasks [80]. Regarding substance use disorders, the Met allele is generally associated with an increased risk of abuse [65,80,81,82], with only one study indicating a protective effect for this allele [66]. Of note, none of these studies adopted BIS-11 to assess impulsivity. This could limit the comparability between studies, since different measures of impulsivity (e.g., self-report questionnaires and behavioral tasks) have been often shown to be poorly convergent [83]. Moreover, none of the studies adjusted the effect of the genotype for educational level or Raven IQ estimates, that in our model were independently associated with impulsivity scores.

### 4.3. HTTLPR x Gender Interaction Effect on Impulsivity Measures

We found that the effect of 5-HTTLPR polymorphism on impulsivity was dependent on gender, with the SS genotype associated with higher impulsivity levels across all the BIS-1 factors in females, but not in males.

The relationship between 5-HTTLPR status and serotoninergic brain activity is unclear. While the S allele has been traditionally associated with higher serotoninergic activity, a more complex picture emerges from recent studies, in which the SNP effect varies when different serotonin receptors are considered [84,85,86]. Regarding the relationship between 5-HTTLPR gender and serotoninergic activity, the SS genotype was found to be associated with lower 5-Hydroxyindoleacetic acid (5-HIAA) levels in men, but higher 5-HIAA levels in females [87]. If this assumption holds true also for our sample, it may suggest that, in females, higher central serotoninergic tone is associated with higher impulsivity. In support of this, it was previously shown how acute tryptophan depletion, a model simulating a low serotoninergic tone, increases impulsivity in males but decreases it in females [88].

Although a study measuring impulsivity through BIS-11 found no interactions between the 5-HTTLPR status and impulsivity scores [89], several other reports are in line with our result of higher impulsivity in females carrying the S allele. In healthy females, the SS genotype was found to be associated with deficits in working memory [90], which is closely dependent on Attentional processes [91]. Another model of Attentional and Motor impulsivity is represented by dysregulated eating behavior [92,93]. In women samples, the 5-HTTLPR S allele increased the risk of eating disturbances [94] and was associated with higher impulsivity in those prone to binge eating [95]. Furthermore, the SS genotype has been shown to be associated with higher disinhibition in adolescent girls [96], as well as hostility and aggressive behaviors [97], features that may be interpreted as surrogates of Motor impulsivity [98]. However, when suicidal behavior is considered as a surrogate measure of impulsivity, opposite results seem to emerge. Indeed, previous studies found an increased risk of suicide risk [99] and higher impulsivity in suicide attempts [100] among men carrying the S allele, as well as higher suicidal behavior in women with the LL genotype [101]. This could be the result of various factors. First, we cannot control for the effect of diagnosis of suicidal behavior, since in our sample data on suicidal behavior history were not available. Second, suicide risk may not necessarily represent a reliable surrogate of impulsivity, since only some types of suicidal behavior seem to be associated with impulsivity [102]; however, previous evidence found BIS-11 scores to be higher among suicide attempters [103,104].

Our results should be interpreted considering some limitations. First, we considered the effect of only three SNPs on impulsivity levels. This surely represents the main limitation of our study, considering the well-known pitfalls of candidate gene studies, such as being unpowered to detect effects of specific variants on genetically complex traits such as impulsivity [105]. Moreover, candidate gene-by-environment interaction findings often suffer from the “replication crisis” phenomenon [106]. Still, these three polymorphisms have been shown to significantly impact various neuropsychological and psychopathological dimensions among various disorders. Second, it is possible that a modulating effect of the three SNPs studied may have been hidden by the relatively small sample size employed in our study; however, the numerosity of our sample was comparable to the majority of other studies with similar design [26,52,107]. Third, the two groups had significant differences in terms of sociodemographic variables; however, although the mixed regression models accounted for these differences, an effect on our findings cannot be completely ruled out. Since impulsivity is a multidimensional and likely multifactorial trait, it is likely to be influenced by many clinical and non-clinical variables; in this context, a significant number of predictors of impulsivity changes in our sample could have been missed due to practical concerns. Finally, BIS is a self-administered questionnaire, with the intrinsic limitations of a subjective assessment. Other neuropsychological tests can be used to objectively assess the impulsivity trait, such as the Go/No Go, the Continuous Performance Test, the Stop Signal Task and the Iowa Gambling Task. However, the BIS is a validated instrument, and it is among the most widely adopted instruments for the assessment of impulsivity among clinical and non-clinical populations.

## 5. Conclusions

In our sample of BD individuals and HC, we found impulsivity levels, as measured by the BIS instrument, to be independently influenced by the *BDNF* rs6265 polymorphism. In addition, an interaction between the 5-HTTLPR status and gender was found, with the SS genotype associated with higher impulsivity levels among females. Of note, BD diagnosis did not exert any significant influence on the relationship between SNP status and impulsivity scores, suggesting a comparable effect of the polymorphisms among HC and individuals with BD.

Impulsivity represents a trait-like feature of BD and has been demonstrated to have prognostic implications in the disease; however, only few studies so far have explored the contribution of the genetic component in modulating impulsivity levels in these patients. Since impulsivity is a complex and multidimensional trait, studies employing new generation sequencing techniques, together with big data analysis methods, are needed to better define the genetic basis of trait impulsivity in BD.

## Figures and Tables

**Figure 1 genes-13-00482-f001:**
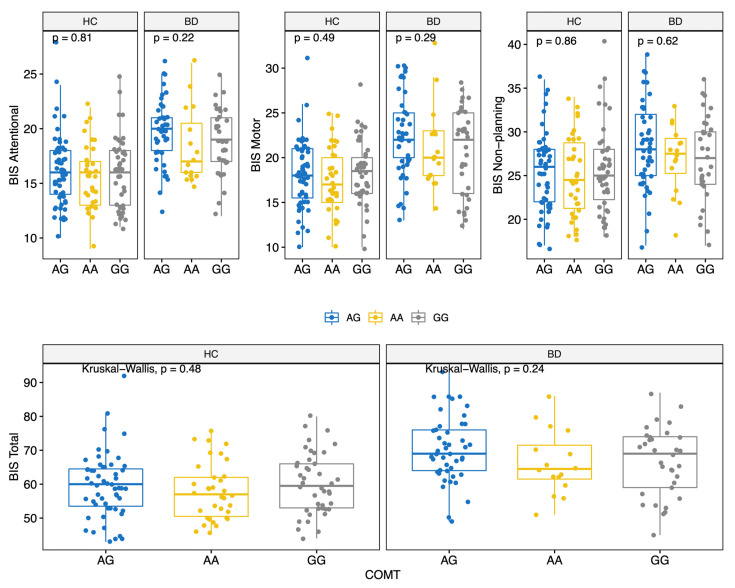
BIS-11 score (Attentional, Motor, Non-Planning and Total) by COMT polymorphisms for HC and BD.

**Figure 2 genes-13-00482-f002:**
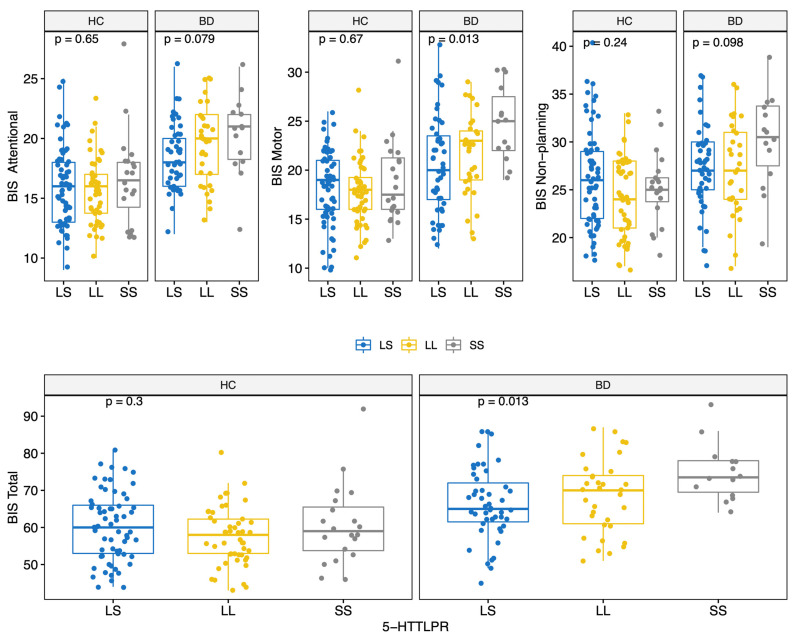
BIS-11 score (Attentional, Motor, Non-Planning and Total) by 5-HTTLPR polymorphisms for HC and BD.

**Figure 3 genes-13-00482-f003:**
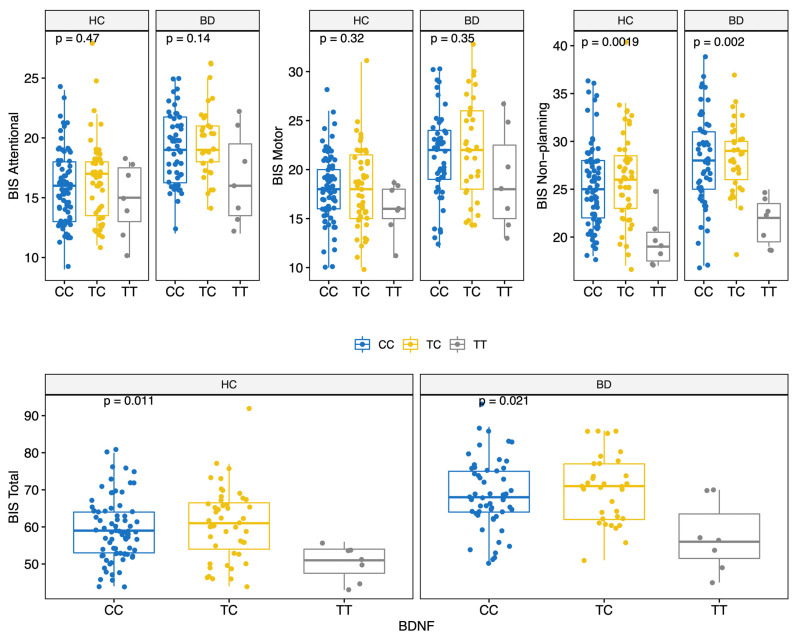
BIS score (Attentional, Motor, Non-Planning and Total) by BDNF polymorphisms for HC and BD.

**Figure 4 genes-13-00482-f004:**
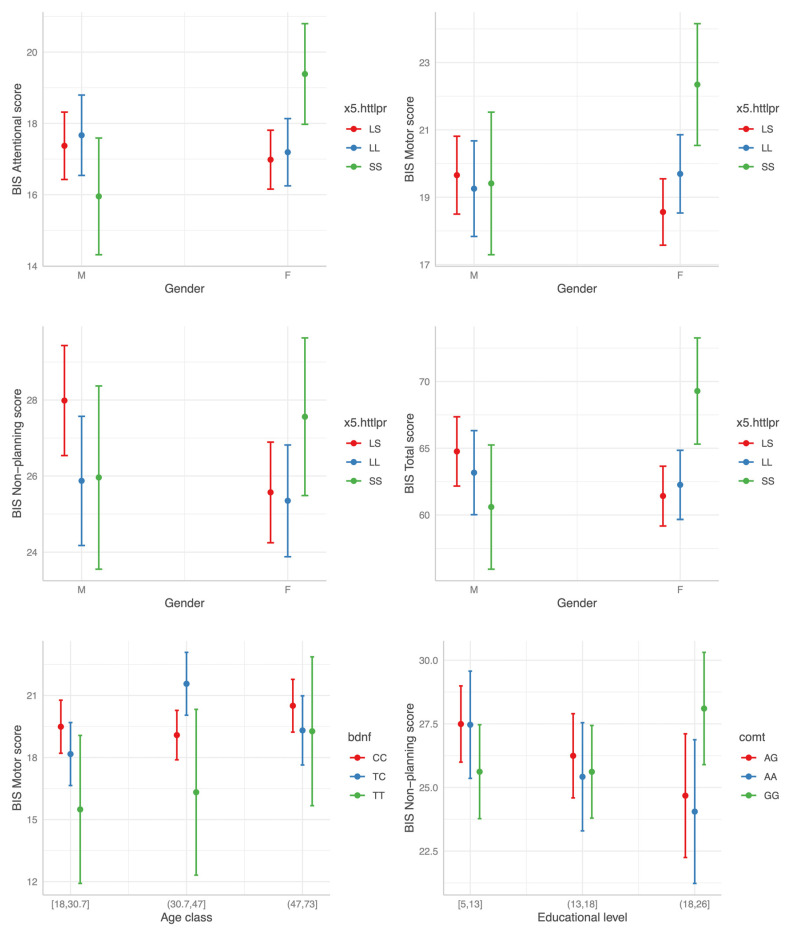
The interaction effect between polymorphism status and gender, age and educational level (from top left to bottom right) on BIS scores.

**Table 1 genes-13-00482-t001:** Main characteristics of the subjects by case-control status variable.

	Overall (n = 225)	HC (n = 131)	BD (n = 94)	*p*-Value
**Center of enrollment**, n (%)				<0.001
Bologna	5 (2.2)	4 (3.1)	1 (1.1)	
Brescia	35 (15.6)	18 (13.7)	17 (18.1)	
Casale Monferrato	17 (7.6)	2 (1.5)	15 (16.0)	
Milan	37 (16.4)	16 (12.2)	21 (22.3)	
Pavia	53 (23.6)	38 (29.0)	15 (16.0)	
Saronno	19 (8.4)	8 (6.1)	11 (11.7)	
Udine	10 (4.4)	5 (3.8)	5 (5.3)	
Verona	49 (21.8)	40 (30.5)	9 (9.6)	
**Age, years**, Average (SD)	39.8 (13.4)	35.5 (12.9)	45.7 (11.7)	<0.001
**Gender**, n (%)				0.24
Males	92 (40.9)	49 (37.4)	43 (45.7)	
**Raven IQ scale**, Average (SD)	118.4 (11.2)	122.7 (7.9)	112.5 (12.4)	<0.001
**Educational level (years)**, Average (SD)	15.2 (4.4)	16.9 (3.8)	12.7 (4.0)	<0.001
**Socioeconomic status**, Average (SD)	39.0 (15.5)	44.5 (12.9)	31.2 (15.7)	<0.001
**HAM-D** (n = 85), Average (SD)	5.68 (3.80)	-	5.68 (3.80)	
**BRMRS** (n = 88), Average (SD)	4.31 (4.76)	-	4.31 (4.76)	

BD: bipolar disorder; BRMRS: Bech–Rafaelsen manic rating scale; HAM-D: Hamilton rating scale for depression; HC: healthy controls.

**Table 2 genes-13-00482-t002:** BIS-11 scores and polymorphism distribution by case-control status variable.

	Overall (n = 225)	HC (n = 131)	BD (n = 94)	*p*-Value
**BIS-11 score**, Median (IQR)				
Attentional	17.0 (5.0)	16.0 (5.0)	19.0 (4.0)	<0.001
Motor	19.0 (6.0)	18.0 (5.0)	22.0 (7.0)	<0.001
Non-Planning	26.0 (6.0)	25.0 (6.0)	28.0 (6.5)	<0.001
Total	63.0 (14.0)	59.0 (12.0)	68.5 (12.8)	<0.001
**COMT Genotype**, n (%)				0.28
AA	50 (22.2)	34 (26.0)	16 (17.0)	
AG	100 (44.4)	55 (42.0)	45 (47.9)	
GG	75 (33.3)	42 (32.1)	33 (35.1)	
**COMT Allele**, n (%)				0.21
A	200 (44.4)	123 (46.9)	77 (41.0)	
G	250 (55.6)	139 (53.1)	111 (59.0)	
**5-HTTLPR Genotype**, n (%)				0.96
LL	81 (36.0)	48 (36.6)	33 (35.1)	
LS	110 (48.9)	63 (48.1)	47 (50.0)	
SS	34 (15.1)	20 (15.3)	14 (14.9)	
**5-HTTLPR Allele**, n (%)				0.90
L	272 (60.4)	159 (60.7)	113 (60.1)	
S	178 (39.6)	103 (39.3)	75 (39.9)	
**BDNF Genotype**, n (%)				0.81
CC	131 (58.2)	77 (58.8)	54 (57.4)	
TC	80 (35.6)	47 (35.9)	33 (35.1)	
TT	14 (6.2)	7 (5.3)	7 (7.4)	
**BDNF Allele**, n (%)				0.67
C	342 (76.0)	201 (76.7)	141 (75.0)	
T	108 (24.0)	61 (23.3)	47 (25.0)	

BD: bipolar disorder; BIS-11: Barratt impulsiveness scale; HC: healthy control.

**Table 3 genes-13-00482-t003:** Coefficients (β) estimated by the mixed-effects regression model considering the Total BIS score as dependent variable.

	BIS Total		
Predictors	β	95% CI	*p*-Value
(Intercept)	65.7	61.1–70.3	**<0.001**
Disease (BD)	5.35	2.40–8.30	**<0.001**
Gender (Female)	−3.34	−6.6–0.01	**0.050**
Age class (31.7–47)	2.22	−0.84–5.27	0.154
Age class (48–73)	2.16	−1.17–5.49	0.202
Raven IQ scale (115–127)	−2.68	−5.58–0.23	0.071
Raven IQ scale (128–128)	−1.80	−5.62–2.02	0.355
Educational level (14–18)	−2.91	−5.87–0.06	0.054
Educational level (19–26)	−4.00	−7.86–−0.14	**0.043**
COMT (AA)	−2.29	−5.31–0.73	0.136
COMT (GG)	−0.88	−3.57–1.82	0.522
5-HTTLPR (LL)	−1.59	−5.57–2.39	0.431
5-HTTLPR (SS)	−4.16	−9.42–1.10	0.120
BDNF (TC)	0.86	−1.65–3.36	0.501
BDNF (TT)	−10.2	−15.2–−5.31	**<0.001**
5-HTTLPR (LL) × Gender (F)	2.43	−2.77–7.63	0.359
5-HTTLPR (SS) × Gender (F)	12.0	5.16–18.9	**0.001**
σ^2^	74.08		
τ_00_	0.54		
ICC	0.01		
N	8		
Observations	225		
Marginal R^2^/Conditional R^2^	0.331/0.336		

Reference category: disease (HC), gender (male), age class (<31.7), Raven IQ scale (<115), educational level (<14), *COMT* (AG), 5-HTTLPR (LS), *BDNF* (CC).

## Data Availability

Authors agree to make data and materials supporting the results or analyses presented in this paper available upon reasonable request.

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
