# Peer review of "The Influence of 5-HTTLPR, BDNF Rs6265 and COMT Rs4680 Polymorphisms on Impulsivity in Bipolar Disorder: The Role of Gender"

_genes, 2022, doi:10.3390/genes13030482_

Round 1

Reviewer 1 Report

This article by Boscutti and colleagues is a well written, comprehensive review on the genetic determinants of impulsivity in bipolar disorder. In this manuscript, Boscutti and coworkers have recruited Individuals with BD and Healthy Controls (HC) in the context of an observational, multisite, study (GECO-BIP). Subjects were genotyped for three candidate polymorphisms (5-HTTLPR, COMT rs4680, and BDNF rs6265); impulsivity was measured through the Italian version of the Barratt Impulsiveness Scale (BIS-11). They built a mixed-effects regression model, with BIS scores as dependent variables, genotypes of the three polymorphisms as fixed effects, and centers of enrollment as random effect. They found that compared to HC, scores for all BIS factors were higher among subjects with euthymic BD (adjusted β for total BIS score: 5.35, p < 0.001). Additionally, they found no significant interaction between disease status (HC vs BD) and SNPs status for any polymorphism. Considering the whole sample, BDNF Met/Met homozygosis was associated with lower BIS scores across all three factors (adjusted β for total BIS score: -10.2, p < 0.001). A significant 5-HTTLPR x gender interaction was found for the SS genotype, associated with higher BIS scores in females only (adjusted β for total BIS score: 12.0, p = 0.001). Finally, COMT polymorphism status was not significantly associated with BIS scores. The authors concluded that BD diagnosis did not influence the effect on impulsivity scores for any of the three SNPs considered. Only one SNP – the BDNF rs6265 Met/Met homozygosis - was independently associated with lower impulsivity scores. The 5-HTTLPR SS genotype was associated with higher impulsivity scores in females only. They stated that there is a need of further studies in adopting genome-wide screening in larger samples to define the genetic basis of impulsivity in BD.

The contents of this review article is well written, comprehensive and  lies within the aims and scope of the Journal ‘Genes’, under the molecular genetics and genomics section of the special issue ‘The relationship between psychiatric disorders and genetics’ , as it highlights the correlation of genetic determinants associated with triggering impulsivity in bipolar disorder.

Reviewer 2 Report

The Article entitled Genetic variants interact with gender to modulate impulsivity in bipolar
disorder: a study on, BDNF rs6265 and COMT rs4680 polymorphisms”, submitted by Boscutti and
Pigoni et al., investigates the effects of three SNPs on impulsivity in individuals with bipolar
disorders, in comparison to healthy controls. Among the genetic variants, the best characterized is
the 5-HTTLPR, for which authors report a correlation of the SS genotype with an increased level of
impulsivity only in female patients. Although no correlation with gender has been found in the
studies on BDNF and COMT, I think the work is well done and clearly described. However, the title
of the paper results to be a bit confusing a not comprehensive of the whole project.

I would suggest to change it.

Below, I report some minor revisions.

In addition, I suggest a moderate text editing and English revision.

Minor Revisions

Line 34: Please, change “Healthy Controls” in healthy controls, as reported in line 132.

Line 39: Correct “al” in “all”.

Lines 86 and 90: Correct “details” in “detail”.

Line 91: In order to avoid repetition, change the sentence “More in detail” appropriately.

Line 94: Number of codon is reported as a reference number. Please, adjust the format.

Line 102: Remove comma in the sentence.

Line 114: What BD II patients are?

Line 134: “The samples... are...” is more appropriate in my opinion.

Line 220: 15.6 corresponds to 15.7 in the table. Please, choose a unique value in both text and
table.

Line 252: 6.1 corresponds to 6.2 in the table. Please, choose a unique value in both text and table.

...and carefully check all the text.

In some lines, references are reported in a confusing manner. Please, correct the format.
